



# FORest Canopy Atmosphere Transfer (FORCAsT) 2.0: model updates and evaluation with observations at a mixed forest site

Dandan Wei[1], Hariprasad D. Alwe[2], Dylan B. Millet[2], Brandon Bottorff[3a], Michelle Lew[3a], Philip S. Stevens[3b], Joshua D. Shutter[4], Joshua L. Cox[4], Frank N. Keutsch[4], Qianwen Shi[5], Sarah C. Kavassalis[5], Jennifer G. Murphy[5], Krystal T. Vasquez[6a], Hannah M. Allen[6a], Eric Praske[6a], John D. Crounse[6b], Paul O. Wennberg[6c], Paul B. Shepson[7], Alexander A.T. Bui[8], Henry W. Wallace[8], Robert J. Griffin[8], Nathaniel W. May[9], Megan Connor[9], Jonathan H. Slade[9], Kerri A. Pratt[9], Ezra C. Wood[10], Mathew Rollings[11,*], Benjamin L. Deming[11,**], Daniel C. Anderson[10,***], and Allison L. Steiner[1]

[1]Department of Climate and Space Sciences and Engineering, University of Michigan, Ann Arbor, MI, USA
[2]Department of Soil, Water and Climate, University of Minnesota, Twin Cities, St. Paul, MN, USA
[3a]Department of Chemistry, Indiana University, Bloomington, IN, USA
[3b]School of Public and Environmental Affairs, Indiana University, Bloomington, IN, USA
[4]Department of Chemistry and Chemical Biology, Harvard University, Cambridge, MA, USA
[5]Department of Chemistry, University of Toronto, Toronto, Ontario, Canada
[6a]Division of Chemistry and Chemical Engineering, California Institute of Technology, Pasadena, CA, USA
[6b]Division of Geological and Planetary Sciences, California Institute of Technology, Pasadena, CA, USA
[6c]Divisions of Engineering and Applied Science and Geological and Planetary Science, California Institute of Technology, Pasadena, CA, USA
[7]Department of Chemistry, Purdue University, West Lafayette, IN, USA and School of Marine and Atmospheric Sciences, Stony Brook University, Stony Brook, NY, USA
[8]Department of Civil and Environmental Engineering, Rice University, Houston, TX, USA
[9]Department of Chemistry, University of Michigan, Ann Arbor, MI, USA
[10]Department of Chemistry, Drexel University, Philadelphia, PA, USA
[11]Department of Chemistry, University of Massachusetts, Amherst, MA, USA
*now at: Department of Chemistry, University of California, Berkeley, CA, USA
**now at: Department of Chemistry and Cooperative Institute for Research in Environmental Sciences, University of Colorado, Boulder, CO, USA
***now at: Universities Space Research Association, Columbia, MD and NASA Goddard Space Flight Center, Greenbelt, MD USA

**Correspondence:** Dandan Wei (dandanwe@umich.edu)

**Abstract.** The FORCAsT (FORest Canopy Atmosphere Transfer) model version 1.0 is updated to FORCAsT 2.0 by implementing five major changes, including (1) a change to the operator splitting, separating chemistry from emission and dry deposition, which reduces the run time of the gas-phase chemistry by 70% and produces a more realistic in-canopy profile for isoprene; (2) a modification of the eddy diffusivity parameterization to produce greater and more realistic vertical mixing in the boundary layer, which ameliorates the unrealistic simulated end-of-day peaks in isoprene under well-mixed conditions and improves daytime air temperature; (3) updates to dry deposition velocities with available measurements; (4) implementation of the Reduced Caltech isoprene mechanism (RCIM) to reflect the current knowledge of isoprene oxidation; and (5) extension of the aerosol module to include isoprene-derived aerosol (iSOA) formation. Along with the operator splitting, modified ver-





tical mixing and dry deposition, RCIM improves the estimation of first generation isoprene oxidation products (methyl vinyl
ketone and methacrolein) and some second generation products (such as isoprene epoxydiols). Inclusion of isoprene in the
aerosol module in FORCAsT 2.0 leads to a 7% mass yield of iSOA. The most important iSOA precursors are IEPOX and
tetrafunctionals, which together account for >86% of total iSOA. The iSOA formed from organic nitrates are more important
in the canopy, accounting for 11% of the total iSOA. The tetrafunctionals compose up to 23% of the total iSOA formation,
highlighting the importance of the fate (i.e. dry deposition and gas-phase chemistry) of later-generation isoprene oxidation
products in estimating iSOA formation.

## 1  Introduction

Forests cover 30% of the land surface and play an important role in the Earth system through exchanges of energy, water, carbon
dioxide, and reactive chemical species with the atmosphere (Bonan, 2008). Forest canopies emit large amounts of volatile
organic compounds (VOCs) into the atmosphere (Guenther et al., 2006) that drive atmospheric chemistry (e.g., Chameides
et al., 1992; Taraborrelli et al., 2012) and are precursors to climate-relevant species such as ozone (e.g., Wolfe et al., 2011) and
particulate matter (e.g., Palm et al., 2016). In addition, forest canopies serve as a major sink of VOCs through dry deposition
(e.g., Nguyen et al., 2015). These bi-directional exchanges and their influences on atmospheric chemistry are complicated by
the three-dimensional structure of the forest canopies, which creates turbulent flows significantly different from the overlying
atmospheric boundary layer (e.g., Gao et al., 1993; Patton et al., 2001) and affects the vertical transport and the chemistry of
trace gases (e.g., Kaser et al., 2015).

The complexity and interplay of these chemical and physical processes challenge our understanding of forest-driven climate
impacts on local, regional, and global scales (e.g., Spracklen et al., 2008; Vanwalleghem and Meentemeyer, 2009). Improving
our understanding of the chemical and physical processes governing the forest-atmosphere interactions at a local scale is
helpful to generalize the net impact of the terrestrial biosphere on chemistry and climate at broader scales. Canopy-chemistry
models that explicitly represent the physical, chemical, and biological processes of an individual forest canopy are useful tools
to investigate the chemically-relevant interactions between forests and the atmosphere at local scales. These canopy-chemistry
models calculate the environmental variables that drive emissions, dry deposition, turbulent mixing, and chemical reactions
vertically throughout a canopy at very fine resolutions (e.g., on the order of meters), while atmospheric chemical transport
models approximate canopy processes through parameterizations and operate on the 10–200 km scale. Several forest canopy-
chemistry models (Stroud, 2005; Forkel et al., 2006; Boy et al., 2011; Wolfe and Thornton, 2011) have been developed to
study the chemically-relevant forest-atmosphere exchanges with the focus on the gas-phase chemical processes. The FORCAsT
model version 1.0 (FORest Canopy Atmosphere Transfer) (Ashworth et al., 2015) is one of the few canopy models currently
capable of simulating the formation of secondary organic aerosols (SOA) from biogenic VOC oxidation.

Over the past decade, the understanding of isoprene chemistry under a wide range of $NO_x$ conditions and their impact
on atmospheric particles has greatly expanded. Specifically, understanding of isoprene oxidation under low-$NO_x$ conditions
has improved (Wennberg et al., 2018), and proper representation of isoprene oxidation and isoprene-derived SOA formation in



canopy-chemistry models is now recognized to be important for a more accurate understanding of forest-atmosphere exchange. Isoprene-hydroxy-peroxy radicals (ISOPOO), produced by addition of a hydroxyl radical (OH) across one of the double bonds followed by the rapid addition of molecular oxygen ($O_2$), react with nitric oxide (NO), hydroperoxy radicals ($HO_2$), other

peroxy radicals, or undergo unimolecular isomerization. Historically, the dominant fate of ISOPOO was thought to be reaction with NO, as mechanisms were developed for urban locations and the NO loss pathway dominates in polluted regions. Under low-$NO_x$ conditions common in forested regions, unimolecular chemistry and reaction with $HO_2$ are also important. The first generation product of the reaction of ISOPOO with $HO_2$, hydroxy hydroperoxide (ISOPOOH), is an important SOA precursor following their oxidation to epoxydiol products (Surratt et al., 2006; Paulot et al., 2009). The other novel low-$NO_x$

pathway recently elucidated is ISOPOO isomerization, which can sustain elevated OH concentrations under low-NO conditions (Peeters and Müller, 2010; Crounse et al., 2011; Teng et al., 2017; Møller et al., 2019). These different branches of ISOPOO pathways produce a different ensemble of oxygenated compounds with low volatility and thus are crucial for accurate prediction of the environmental and climate impacts of isoprene chemistry. In addition to the gas-phase fate of isoprene, field studies found evidence of C5 compounds in ambient particles (Claeys, 2004; Kleindienst et al., 2007), and the modeling of isoprene-

derived SOA has been significantly advanced in the past decade (e.g., Compernolle et al., 2009; Marais et al., 2016; Gaston et al., 2014). Wennberg et al. (2018) compiles a comprehensive isoprene mechanism incorporating the current knowledge of isoprene chemistry and includes the necessary isoprene SOA precursors. The Reduced Caltech Isoprene Mechanism (RCIM), a reduced version with the same product yields of known compounds and minimal simplifications beyond lumping of isomeric compounds and removal of minor (< 2% yield) pathways, is developed concurrently with the explicit mechanism and is more

suitable for implementing in canopy-chemistry models.

Here, we develop FORCAsT 1.0 to FORCAsT 2.0 to incorporate important updates to gas-phase isoprene chemistry, isoprene-derived SOA formation, and two physical components of the model. Specifically, major updates include (i) separating the integration of chemistry from the emission and dry deposition (a.k.a. operator splitting) to provide more realistic representations of vertical gradients in the forest canopy and to make the chemical module more flexible with future chemical

mechanism updates; (ii) improving the vertical mixing parameterization in the boundary layer; (iii) updating the dry deposition velocities for chemical species with available measurements (Nguyen et al., 2015); (iv) implementing the RCIM to reflect the current understanding of isoprene fate under low $NO_x$ conditions (Wennberg et al., 2018); and (v) extending the MPMPO aerosol module (Griffin et al., 2005; Ashworth et al., 2015) to include isoprene-derived SOA formation. We evaluate FORCAsT 2.0's performance against FORCAsT 1.0 (Ashworth et al., 2015) and the observations from the AMOS (Atmospheric

Measurements of Oxidants in Summer) field campaign, conducted at the University of Michigan Biological Station (UMBS) during the summer of 2016.

## 2 Model description

The FORCAsT model, based on the CACHE canopy model by Forkel et al. (2006), is a one dimensional model that couples atmospheric chemistry (gas-phase and gas-particle partitioning) and canopy processes. The vertical resolution of FORCAsT





**Table 1.** The order of operation in the FORCAsT model. $\frac{\partial C_i}{\partial t}$ denotes the time evolution of concentrations of chemical species $i$.

| FORCAsT 1.0 | FORCAsT 2.0 |
|---|---|
| $\frac{\partial C_i}{\partial t}$ = transport | $\frac{\partial C_i}{\partial t}$ = transport + emission + dry deposition |
| ⇓ | ⇓ |
| $\frac{\partial C_i}{\partial t}$ = chemistry + emission + dry deposition | $\frac{\partial C_i}{\partial t}$ = chemistry |

can be configured with a minimum of 20 and a maximum of 60 vertical layers. In this study, the number of model levels is set to 40, with 20 layers within the canopy, 8 layers representing the boundary layer ($\sim$1 km), and the remaining layers extending to the lower troposphere ($\sim$4 km). In addition to the above-ground layers, the model includes 15 soil layers for computing soil heat and moisture storage, as well as exchange with the atmosphere and root extraction (Forkel et al., 2006).

  The FORCAsT model parameterizes the processes of radiative transfer, chemical species emission, advection, deposition,
vertical exchange, and chemistry, and then integrates the energy and mass balance equations at each vertical layer in the canopy. The parameterization of the radiative transfer, emission, advection, turbulent mixing in the canopy layers has been described extensively in Bryan et al. (2012) and Ashworth et al. (2015) and remain unchanged in the updated version 2.0. Here we describe the major updates to the model, including the operator-splitting, gas-phase chemistry, gas-particle partitioning, and some aspects of the dry deposition and turbulent mixing.

## 2.1 Operator splitting

Processes such as emission, turbulent mixing, dry deposition, and chemical reactions occur at the same time in the atmosphere. However, in numerical models, these processes (i.e. operators) are split and integrated over time and/or space in sequence, commonly referred to as operator splitting. It is generally faster to integrate the operators separately than to compute the solution when the operators are treated together (Lapointe et al., 2020). This computational efficiency comes at the cost of an
error introduced by the splitting. In the context of atmospheric chemistry modeling, model accuracy is affected by the order in which the operators are applied (Santillana et al., 2016), and by the integration timesteps of the operators (a.k.a operator duration) (Philip et al., 2016). In prior studies of chemical transport models, the operator duration causes greater differences in concentrations of reactive emitted species such as nitrogen oxides (up to 5 times, Philip et al., 2016) than the order of operators (up to 10%, Santillana et al., 2016).
In FORCAsT 1.0, the order of operation is from vertical transport to chemistry, with emissions and dry deposition integrated within the chemistry solver (Table 1). The chemistry solver typically dominates the computational cost of the simulations (Lapointe et al., 2020). To increase computational efficiency as well as to allow for flexible chemical mechanisms in the future, we separate the chemical solver from emission and dry deposition and integrate the latter two operators with the vertical transport (Table 1). The impacts of this operator-splitting on FORCAsT 2.0's performance are discussed in section 3.1.





## 2.2 Turbulent mixing

In the surface layer (roughly 10% of the boudary layer), the eddy diffusivity K is commonly defined as a simple function of height $z$ of the form $K = \kappa u_* z$, where $\kappa$ is the von Karman constant and $u_*$ is the friction velocity (Stull, 1988). In the rest of the boundary layer, defining K is not as clear for numerical models. Several approaches have been used in the literature to define the K profile, such as linearly decreasing from the surface layer to top of the boundary layer (Estoque, 1963), exponentially decreasing with height from the surface layer, and by finding an interpolating polynomial passing through prescribed points with predefined slopes (O'Brien, 1970). The general approach for approximating K in the boundary layer has used a power law dependence on $z/z_i$, where $z_i$ is the top of the boundary layer and scale parameters were derived from similarity theory or empirically. A commonly used shape function is of the form (Troen and Mahrt, 1986):

$$K = \kappa u_* z (1 - \frac{z}{z_i})^p \tag{1}$$

The exponent p = 2 has been commonly used in the literature (Nissanka et al., 2018), while values between 2 and 3 agree with different observed profiles.

The eddy diffusivity (K) parameterization based on mixing-length theory in FORCAST 1.0 is described as follows, with greater detail presented in Forkel et al. (2006) and Bryan et al. (2012):

$$K = l^2 \frac{\partial u}{\partial z} \tag{2}$$

$$l = \frac{\kappa z}{1 + \frac{\kappa z}{\lambda}} \tag{3}$$

$$\frac{\partial u}{\partial z} = \frac{u_*}{\kappa z} \tag{4}$$

where $l$ is the mixing length, $u$ is the mean wind speed, $\kappa$ is the von Karman constant (0.41), $z$ is the height above the ground, and $u_*$ is the friction velocity. The $\frac{\partial u}{\partial z}$ is derived from the common logarithmic expression for the boundary layer (Equation 4). Combing Equation (2), (3) and (4) and rearranging the terms:

$$K = \kappa u_* \frac{z}{(1 + \frac{\kappa z}{\lambda})^2} \tag{5}$$

where $\lambda$ is a function of the height $z$ as shown below:

$$\lambda = \begin{cases} 2.0, & z < h_c \\ max(0.1z_i,\ 2.7 \times 10^{-4} \frac{G}{f}), & h_c \leqslant z \leqslant z_i \\ 2.7 \times 10^{-4} \frac{G}{f}, & z \geqslant z_i \end{cases} \tag{6}$$





where $h_c$ is the canopy height, $z_i$ is the height of the boundary layer, G is the geostrophic wind at the top of the boundary layer and set to 17 m s$^{-1}$, and $f$ is the Coriolis parameter. This parameterization yields a relatively small value of K (i.e. <10 m$^2$s$^{-1}$) in the boundary layer and thus implicitly produces a low boundary layer height ($\sim 250-300$ m around 14:00 local time) in FORCAST 1.0. In addition, this mixing parameterization in FORCAST 1.0 produces an unrealistic end-of-day peak in isoprene under well-mixed conditions.

To produce more realistic K in the boundary layer and boundary layer height, we adopt the parameterization described in Equation 1 to calculate the K in the boundary layer. In the present study, we use a cubic power of height z (i.e., p = 2) for K in the boundary layer:

$$
\mathrm{K_{new}} = \begin{cases} \frac{\kappa u_* z}{\phi(z/L)}, & z < z_{\mathrm{sfc}} \\ \frac{a\kappa u_* z (1-\frac{z}{z_i})^2}{\phi(z/L)}, & z \geqslant z_{\mathrm{sfc}} \end{cases} \tag{7}
$$

where $z_{\mathrm{sfc}}$ is the surface layer height (here assumed to be 10% of the boundary layer height $z_i$), and the constant a = $1/(1-\frac{z_{\mathrm{sfc}}}{z_i})^2$

= 1.23 is used to ensure a continuous transition of $\mathrm{K_{new}}$ from the surface layer to the boundary layer. The $z_i$ is calculated as a function of the sensible heat flux at the top of the canopy (Stull, 1988):

$$
z_i = \sqrt{\frac{2(2c+1)}{\gamma} \int_0^t \overline{(w'\theta')}_{cpy}} \tag{8}
$$

where c is a standard entrainment parameter (=0.2), and $\gamma$ is the lapse rate in the free atmosphere. The calculation for $z_i$ starts when the heat flux $\overline{(w'\theta')}_{cpy}$ first becomes positive in the morning, and continues one more hour after $\overline{(w'\theta')}_{cpy}$ becomes

negative. The stability function $\phi(z/L)$ is given in equation 9 (Nissanka et al., 2018):

$$
\phi(\mathrm{z/L}) = \begin{cases} (1-16z/L)^{-1/2}, & z/L < 0 \\ 1+5z/L, & z/L > 0 \\ 1, & z/L = 0 \end{cases} \tag{9}
$$

Overall, the new parameterization produces a larger K, a more realistic boundary layer height, and thus a more realistic air temperature (see details in Section 3.2).

### 2.3 Dry deposition

The dry deposition velocity to the canopy foliage ($v_d$) is calculated using the resistance model (Meyers and Baldocchi, 1988; Wesely, 1989) in FORCAST 1.0, with details provided in Bryan et al. (2012) and Ashworth et al. (2015). Recent work by Nguyen et al. (2015) report dry deposition velocities based on measured fluxes and concentrations for 16 atmospheric compounds above a southeastern United States forest, suggesting the parameterization in the FORCAST 1.0 underestimates dry deposition velocities for these oxygenates (Table 2). We adopt this newer parameterization in FORCAST 2.0, where the major

revisions to the resistance model include (1) the addition of the aerodynamic resistance $R_a$ ; (2) the formulation of the molecular diffusion $R_b$; and (3) the addition of temperature dependence to the mesophyll resistance $R_m$ and the cuticular resistance $R_c$.





**Table 2.** Estimates of the dry deposition velocities (cm s$^{-1}$) to the canopy foliage for relevant molecules at 12:00 local time. The measurement-based deposition velocities for a temperate forest are provided as a reference (Nguyen et al., 2015).

| Species | Revised resistance model | Old resistance model | Measurement-based |
|---|---|---|---|
| $H_2O_2$ | 5.2 | 0.2 | 5.0±1.0 |
| $HNO_3$ | 4.6 | 7.7 | 4.0±1.0 |
| HCOOH | 1.6 | 3.3 | 1.0±0.4 |
| HAC | 0.7 | 0.1 | 1.0±0.5 |
| HMHP | 4.7 | 1.0 | 4.0±1.0 |
| IHN | 1.3 | 0.0 | 1.5±0.6 |
| HPALD | 3.2 | $5.5\times10^{-2}$ | 2.0±0.6 |
| ISOPOOH/IEPOX | 3.4 | 1.0 | 3.0±0.6 |
| PROPNN | 2.0 | NA | 2.0±0.6 |
| INP | 1.3 | NA | 1.0±0.6 |

Details of the revised parameterization can be found in Nguyen et al. (2015). We apply this method for species where input data are available, including for $H_2O_2$ (hydrogen peroxide), $HNO_3$ (nitric acid), HCOOH (formic acid), HAC (hydroxyacetone), PROPNN (propanone nitrate), HMHP (hydroxymethyl hydroperoxide), INP (isoprene nitrooxy hydroperoxide), IHN (isoprene

hydroxy nitrates), HPALD (hydroperoxy aldehydes), IEPOX (epoxydiols), and ISOPOOH (isoprene hydroxy hydroperoxides) (Table 2). The revised deposition velocity parameterization is also applied to all the isomers for INP, IHN, ISOPOOH, IEPOX, and HPALD in RCIM. Note that the revised dry deposition scheme did not provide an estimate for methyl vinyl ketone and methacrolein due to lack of data, therefore we use the observation-based estimate (up to 2.4 cm s$^{-1}$, Karl et al., 2010) that is 6 times higher than in FORCAsT 1.0 ($<$0.35 cm s$^{-1}$). The estimates of the dry deposition velocity are then scaled by

the leaf area distribution in the canopy. Species other than those listed above use the old parameterization in FORCAsT 1.0. The comparison of dry deposition velocities between the old and new parameterizations is shown in Table 2, with the new parameterization generally increasing the deposition velocity with the exception of nitric and formic acids. As input data and measurements become available for other species, the revised parameterization can be evaluated against new observations and applied to other species.

### 2.4 Isoprene gas-phase mechanism: the Reduced Caltech Isoprene Mechanism (RCIM)

We replace the chemical reactions for isoprene oxidation in the Caltech Atmospheric Chemistry Mechanism (CACM) in FOR-CAsT 1.0 with RCIM. The original CACM was developed for application to urban conditions (CACM0.0) (Griffin et al., 2002, 2005), and the version incorporated into FORCAsT 1.0 was updated for low-NO$_X$ conditions (Ashworth et al., 2015), here-inafter referred to as CACM1.0 in the following sections. RCIM is version 4.3 of the "Reduced" mechanism in the Wennberg

et al. (2018) mechanism repository (Bates and Wennberg, 2017), including the essential chemistry required to accurately simulate isoprene oxidation under remote conditions in the atmosphere. Compiled concurrently with the full explicit mechanism,





RCIM groups isomers with similar reaction rates and products, and lumps minor pathways (< 2% branching ratio) into the major channels to minimize the number of species and reactions while retaining an accurate description of the oxidative fate of those grouped species and lumped pathways (Wennberg et al., 2018). The reduced mechanism includes 119 species and 221

reactions, in contrast to the 385 species and 810 reactions in the explicit isoprene mechanism in Wennberg et al. (2018) and the 113 reactions for isoprene in CACM1.0.

RCIM treats the initial system of allylic and peroxy radicals formed following the addition of OH to isoprene dynamically. Older mechanisms, including CACM1.0 in FORCAsT 1.0, implicitly used fixed distributions of isoprene-hydroxy-peroxy radicals (ISOPOO) derived from experiments performed under high-NO conditions. Addition of $O_2$ to allylic radicals under

ambient conditions is in fact a reversible process, resulting in a dynamic system with interconversion between the six major ISOPOO isomers (two subgroups of three defined by a common OH position) (Peeters et al., 2009; Teng et al., 2017). Wennberg et al. (2018) represent the reversibility of $O_2$ addition in the explicit mechanism by including 10 species (i.e. 4 allylic radicals and 6 major ISOPOO; see Fig. 3 in Wennberg et al., 2018) and 69 reactions. The reduced mechanism RCIM retains this novel treatment of the ISOPOO system (Wennberg et al., 2018), although it simplifies the 10 species radical system to two major

ISOPOO isomers, i.e. (1-OH,4-OO)-ISOPOO and (4-OH,1-OO)-ISOPOO.

RCIM includes important updates to the formation and fates of isoprene hydroxyl nitrates (IHN) through pressure- and temperature-dependent parameterizations of the branching ratios and a new structure–activity relationship for calculating the formation of nitrates from multifunctional peroxy radicals without measured yields (Wennberg et al., 2018). The dynamic representation of the ISOPOO isomers also contributes to higher production of IHN than previous mechanisms when included in

global models (Bates and Jacob, 2019). In addition, RCIM includes 12 distinct C5 tetrafunctional compounds with unique combinations of functional groups. They are dihydroxy hydroperoxy nitrate, carbonyl hydroxy nitrooxy hydroperoxide, dihydroxy carbonyl nitrate, hydroxy hydroperoxy dialdehyde, hydroxy hydroperoxy dinitrate, dihydroxy dinitrate, carbonyl hydroperoxy-diol, carbonyl hydroxy dinitrate, dihydroxy hydroperxy epoxide, dihydroxy dihydroperoxide, hydroxy nitrooxy dihydroperoxide, and hydroxy nitrooxy hydroperoxy epoxide. Each C5 tetrafunctional compound represents a variety of isomers. Cham-

ber experiments suggest that multifunctionals contribute to isoprene-derived SOA (iSOA) (Ng et al., 2008; Schwantes et al., 2019). Additional aspects of RCIM relative to CACM1.0 include decreased C5-hydroperoxy-aldehyde (HPALD) yields following the 1,6-H shifts of the Z-ζ-OH-peroxy radicals in Teng et al. (2017), and additional intramolecular H shifts, including rapid peroxy-hydroperoxy shifts, resulting in higher OH recycling under low-NO conditions (Wennberg et al., 2018). These mechanism changes are manifested by changes in gas-phase isoprene oxidation products (Section 3.3).

## 2.5 Isoprene-derived secondary organic aerosol

FORCAsT simulates the partitioning of condensable species into the particle phase using the Model to Predict the Multiphase Partitioning of Organics (MPMPO, Griffin et al., 2005). In FORCAsT 1.0, 99 out of the 300 prognostic species in CACM1.0 are treated as condensable and lumped into 12 surrogate species according to their structures, sources (biogenic or anthropogenic), volatilities, and dissociative capabilities (Ashworth et al., 2015). Specifically, the 12 surrogate groups include 6 anthropogenic

aromatic groups, 4 monoterpene-derived biogenic surrogates, and 1 group composed of non-volatile dimers of multifunctional



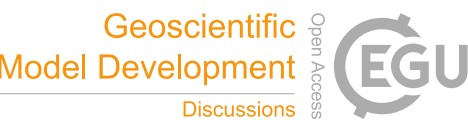

**Table 3.** Properties of the six new iSOA surrogate groups in FORCAsT 2.0. The Henry's law constants below are from Sander (2015) and Safieddine et al. (2017) at 298 Kelvin. Temperature-dependence of Henry's law constants is included in FORCAsT.

| Surrogate species | Representative molar mass (g mol$^{-1}$) | Henry's Law coefficient (at 298K; M atm$^{-1}$) | Surrogate group name |
|---|---|---|---|
| GLYX ($C_2H_2O_2$), MGLY ($C_3H_4O_2$) | 72.0 | $3.24 \times 10^4$ | GLYX |
| IEPOXt, IEPOXc, IEPOXD ($C_5H_{10}O_3$), ICHE ($C_5H_8O_3$), HMML ($C_4H_6O_3$) | 118.0 | $8.0 \times 10^7$ | IEPOX |
| ISOP1N4OH, ISOP1OH4N ISOP1OH2N, ISOP3N4OH ($C_5H_9NO_4$) | 147.0 | $1.74 \times 10^4$ | IHN |
| INPB, INPD ($C_5H_9NO_5$) | 163.0 | $1.74 \times 10^4$ | INP |
| MVK3OOH4N, MACR2OOH3N ($C_4H_7NO_6$) MVK3N4OH, MVK3OH4N ($C_4H_7NO_5$) MACR2OH3N, MACR2N3OH ($C_4H_7NO_5$) | 149.0 | $1.74 \times 10^4$ | C4 |
| C5 tetrafunctional compounds | 208.0 | $1.0 \times 10^8$ | Tetra |

acids from monoterpene oxidation (i.e. phthalic acid). Additionally, one surrogate group based on keto-propanoic acid and oxalic acid (formed from oxidation of methyl vinyl ketone and methacrolein) is considered condensable in CACM1.0-MPMPO. However, explicit formation of SOA from isoprene is missing in FORCAsT 1.0. We incorporate 6 new surrogate groups in MPMPO to account for the isoprene-derived SOA (iSOA) in FORCAsT 2.0 (Table 3).

As the iSOA precursors are small organic molecules (number of carbon atoms $\leq 5$) with numerous functional groups, they are expected to be highly hydrophilic. Under humid conditions representative of the summertime PROPHET boundary layer, aqueous aerosol provides a medium for reactive uptake (Surratt et al., 2009) and thus iSOA are likely aqueous (Couvidat and Seigneur, 2011; Ervens et al., 2011). Marais et al. (2016) proposed a mechanism for irreversible reactive uptake of iSOA precursors by preexisting aqueous aerosols, dependent on the Henry's Law coefficients, that is widely used in chemical transport

models. Bates and Jacob (2019) estimate a global iSOA yield by mass of 25% using this mechanism coupled with RCIM. On the other hand, dry chamber experiments (relative humidity < 10%) suggest up to 11% of iSOA yield (Kroll et al., 2006), suggesting a partitioning between the gas phase and a non-aqueous organic phase. Thornton et al. (2020) estimate an upper bound of the non-aqueous iSOA yield of 3% under atmospheric conditions using a volatility-driven gas-particle partitioning implemented in a box model.

The MPMPO in FORCAsT only considers organic aerosol species and assumes the partitioning of the gases into two particulate phases: a purely organic aerosol and an aqueous aerosol with associated molecular and ionic components. Equilibrium between the gas and aerosols is assumed for organic species. The equilibrium organic aerosol-phase mass concentration of an individual species $i$, $O_i$ ($\mu$g m$^{-3}$ air), is given by the following relationship (Griffin et al., 2005):

$$O_i = K_{om,i} G_i M_o = \frac{G_i M_o RT}{M_{om} 10^6 \gamma_i p_{L,i}^o} \tag{10}$$


where $K_{om,i}$ (m$^3$ air $\mu$g$^{-1}$) is the partitioning coefficient that describes the phase distribution of the condensing organic species (Pankow, 1994), $G_i$ is its corresponding gas phase concentration ($\mu$g m$^{-3}$ air) and $M_o$ is the total concentration ($\mu$g m$^{-3}$ air) of organic aerosol mass available to act as the partitioning medium. R is the ideal gas constant ($8.2 \times 10^{-5}$ m$^3$ atm mol$^{-1}$ K$^{-1}$), T is temperature (K), $M_{om}$ is the average molecular weight (g mol$^{-1}$) of the absorbing organics including both primary and secondary organic compounds, $p^o_{L,i}$ is the pure component vapor pressure (atm) of species $i$, and $\gamma_i$ is the activity coefficient of species $i$ in the organic phase. The factor of $10^6$ converts g to $\mu$g.

The aqueous phase concentration of species $i$ $A_i$ ($\mu$g m$^{-3}$ air) is given by (Griffin et al., 2005):

$$A_i = \frac{G_i \cdot \text{LWC} \cdot \text{H}_i}{\gamma_{aq,i}} \tag{11}$$

where LWC is the aerosol liquid water content ($\mu$g H$_2$O m$^{-3}$ air), $H_i$ is the Henry's law coefficient (($\mu$g $\mu$g$^{-1}$ H$_2$O)/($\mu$g m$^{-3}$ air)), and $\gamma_{aq,i}$ is the activity coefficient of organic species $i$ in the aqueous phase normalized by that at infinite dilution. The aqueous-phase equilibrium is constrained by dissociation of the dissolved organic species, where pH is needed to calculate the concentration of the charged ions (see Equation 13, 14 in Ashworth et al., 2015).

The LWC in Equation 11 is calculated offline as the sum of aerosol liquid water associated with inorganic species (LWC$_{\text{inorg}}$) and organics (LWC$_{\text{org}}$). The LWC$_{\text{inorg}}$ and pH are calculated by the Extended AIM Aerosol Thermodynamics Model model II (E-AIM, http://www.aim.env.uea.ac.uk/aim/model2/model2d.php) (Clegg et al., 1998) using measurements of NH$_3$(g) and PM$_{2.5}$ sulfate, nitrate, and ammonium (Table 4), following the method of Murphy et al. (2017). The LWC$_{\text{org}}$ is calculated according to Petters and Kreidenweis (2007):

$$\text{LWC}_{\text{org}} = \frac{m_{org}\rho_w}{\rho_{org}} \frac{\kappa_{org}}{\frac{1}{\text{RH}} - 1} \tag{12}$$

where $m_{org}$ is the organic mass concentration from HR-ToF-AMS measurements, $\rho_{org}$ is the density of organics (1.4 g cm$^{-3}$; Cerully et al., 2015), $\rho_w$ is the density of water, RH is the relative humidity, and $\kappa_{org}$ is the hygroscopicity growth for organics taken as 0.08 in this study. The $\kappa_{org}$ is derived from the HR-ToF-AMS data using the method described in Cerully et al. (2015):

$$\kappa = \epsilon_{org}\kappa_{org} + \epsilon_{inorg}\kappa_{inorg} \tag{13}$$

where $\epsilon_{org}$ and $\epsilon_{inorg}$ are the volume fractions of organic and inorganic species calculated using AMS mass fraction data, and $\kappa_{inorg}$ is the hygroscopicity growth determined from the speciated inorganic concentrations and $\kappa_{inorg}$ for individual inorganic species from Padró et al. (2010). $\kappa$ is the total hygroscopicity growth, and has been shown to be insensitive to location and organic fraction (Padró et al., 2010). Here we use a value of 0.23 for $\kappa$ based on measurements from previous studies (Padró et al., 2010; Cerully et al., 2015). We calculate LWC as the sum of LWC$_{\text{inorg}}$ and LWC$_{\text{org}}$ based on surface observations, and vertically scale LWC by the relative humidity to extrapolate to other model heights.

The thermodynamic model UNIFAC (UNIversal Functional group Activity Coefficient) is used to calculate the activity coefficients $\gamma$ in Equation 10 and 11 (Fredenslund et al., 1975). The standard UNIFAC parameters (e.g. alkane group) are found in Hansen et al. (1991), Balslev and Abildskov (2002), and Wittig et al. (2003). The UNIFAC parameters for the functional





groups of nitrate and hydroperoxide are taken from Compernolle et al. (2009). The missing UNIFAC parameters (e.g. the unknown interaction parameters) for some functional groups (such as nitrate and hydroperoxide) are set to zero, which introduces uncertainties in estimating the activity coefficient for the new SOA surrogate groups.

## 3 Model evaluation with observations

The performance of FORCAsT 2.0 is evaluated against the observations obtained at the PROPHET (Program for Research on Oxidants: Photo-chemistry, Emissions, Transport) tower at UMBS during the 2016 AMOS field campaign. Full details of the PROPHET tower and the UMBS site can be found in Millet et al. (2018). Measurement details and references are listed in Table 4. The results presented in this study are based on a 2-day model simulation for the two sunny days of 22-23 July 2016, where the first day (July 22) is a well-mixed day and the second day (23 July) is relatively stagnant based on micrometeorological analysis (Wei et al., 2020). The 2-day period is relatively hot for the site, with a mean high of 30.0 °C and a mean daily temperature of 23.9 °C, compared to the monthly averages of 25.4 °C and 21.0 °C, respectively. The canopy structure in the model is identical to that used in Bryan et al. (2012) and Ashworth et al. (2015). The input data are based on the measurements during AMOS 2016, with the model driven by observed PAR (Photosynthetically Active Radiation), standard deviation of the vertical velocity ($\sigma_w$), friction velocity ($u_*$) and the calculated aerosol liquid water content. The initial concentrations for the chemical species are taken from the measurements when available, including ozone ($O_3$), nitric oxide (NO), nitrogen dioxide ($NO_2$), formaldehyde ($CH_2O$), methyl vinyl ketone and methacrolein (MVK+MACR). The isoprene emission factor is increased from prior FORCAsT 1.0 studies (Bryan et al., 2012; Ashworth et al., 2015), as the last measurement campaign (i.e. CABINEX 2009) occurred during a relatively cool period and utilized an emission factor roughly a factor of 2-3 lower than previously observed at the site (Pressley, 2005; Unger, 2013). The 2-day period evaluated in this study is warmer, and increasing the isoprene emission factor is consistent with the effects of prior day temperature (Guenther et al., 2006). The gas-phase chemical mechanism in FORCAsT includes explicit treatment of two monoterpene surrogate species: $\alpha$-pinene (APIN) and d-limonene (DLMN). Light-dependence of monoterpene emissions is included in the two-day simulation by changing the synthesis emission factors from 0 to 0.4 nmol m$^{-2}$ s$^{-1}$ for the two monoterpene surrogates, as this has been observed by Ortega et al. (2007) at the site and this emission change improved the agreement with the measured monoterpene concentrations for a more realistic evaluation of the monoterpene-derived SOA. In the following sections, we describe the impact of model changes (Section 2.0) on the physical and chemical environment in the forest canopy during this 2-day period.

### 3.1 Operator splitting

As described in section 2.1, the chemistry is separated from the operators of emission and dry deposition in FORCAsT 2.0 (Table 1). The splitting of chemistry from the emission and dry deposition leads to weaker gradients in the vertical profiles for emitted species such as isoprene (Fig. 1). The old order of operators simulates higher concentrations between 0.7z/h and 1.0z/h where the emission occurs (Fig. 1a-c), likely due to the higher production rates of isoprene resulted from emission (on the order of $10^{-6} - 10^{-5}$ ppbv s$^{-1}$ depending on the leaf area distribution) compared to the reaction rates (on the order of





**Table 4.** 2016 UMBS PROPHET observations utilized in this study for model evaluation, including concentration measurements, measurement heights, instrumental techniques, and references. Chemical compound abbreviations are defined as follows: MVK + MACR (methyl vinyl ketone and methacrolein), $NO_x$ (nitrogen oxides), $O_3$ (ozone ), OH (hydroxyl radical), $HO_2$ (hydroperoxy radical), $RO_x$ (peroxy radical), $CH_2O$ (formaldehyde), ISOPOOH (isoprene hydroxy hydroperoxide), IEPOX (isoprene epoxydiol), IHN (isoprene hydroxy nitrate), MTN (monoterpene hydroxy nitrate). IEPOX-SOA and 91Fac represent the isoprene-derived epoxydiol organic aerosols and monoterpene-derived organic aerosols, respectively. The canopy height is 22.5 m.

| Measurement | Height [m] | Instrumental technique | Reference |
|---|---|---|---|
| Isoprene<br>Monoterpenes<br>MVK + MACR | 34, 21, 17,<br>13, 9, 5 | PTR-QiTOF (Ionicon Analytik, GmbH) | Millet et al. (2018) |
| $NO_x$ | 29 | A dual-channel custom-built chemiluminescence instrument by Air Quality Design Inc. | Geddes and Murphy (2014)<br>Shi et al. (in prep) |
| $O_3$ | 6 | Model 205 (2B Technologies, Inc.) dual-beam UV absorption instrument | |
| OH, $HO_2$ | 32 | Indiana University Laser-Induced Fluorescence-Fluorescence Assay by Gas Expansion (LIF-FAGE) | Dusanter et al. (2009) |
| $RO_x$ | 30 | Ethane CHemical AMPlifier (ECHAMP) technique | Wood et al. (2016) |
| $CH_2O$ | 5, 17, 21,<br>30 | Harvard Fiber Laser-Induced Fluorescence (FILIF) instrument | Hottle et al. (2009)<br>DiGangi et al. (2011)<br>Cazorla et al. (2015) |
| ISOPOOH, IEPOX, IHN | 32 | High Resolution GC Chemical Ionization Mass Spectrometer (GC-HR-ToF-CIMS) | Vasquez et al. (2018) |
| IHN, MTN | 19.5 | Iodide-adduct Chemical Ionization Mass Spectrometer | |
| Ammonia (gas), PM2.5 sulfate, nitrate, and ammonium | 5.5 | Ambient ion monitor-ion chromatography (AIM-IC, model 9000D, URG Corp., Chapel Hill, NC) | Markovic et al. (2012) |
| Organic aerosols, sulfate, nitrate, ammonium | 30 | HR-ToF-AMS (Aerodyne Research Inc., USA) | Bui et al. (in prep) |
| IEPOX-SOA, 91Fac | 30 | HR-ToF-AMS (Aerodyne Research Inc., USA) and positive matrix factorization | Bui et al. (in prep) |





$10^{-8}$ ppbv s$^{-1}$ for isoprene + OH) in the same solver, resulting in the vertical concentration profiles resembling the emission

profiles. The new order of operators simulates more realistic in-canopy concentration gradients for the emitted species such as isoprene. Specifically, the vertical profile of isoprene before sunrise is better captured by the new order of operation with a RMSE (root-mean-square error) less than 0.1 (Fig. 1a, d). The mid-day in-canopy profiles are also improved by the new order, while simulated concentrations are higher above-canopy than observed (Fig. 1b, e). The overall RMSE for the mid-day case (Fig. 1e) suggest the results for the new order agree slightly better with the observations. The differences between the

two orders are up to 16% around the height of 0.85z/h in the mid-day case and are in good agreement with Santillana et al. (2016) who report a 10% difference. In the evening hours, the emission rates and chemistry quickly decrease due to the reduced radiation, leading to more fluctuations in the observed profile that are challenging for models to capture (Fig. 1c).

In addition, the time for running the model with the gas-phase chemistry alone (i.e. without running the aerosol module) is reduced from 10 minutes to 3 minutes. In chemical models, the computational cost is mainly due to the chemical solver which

has a very small internal time step (about 1.2 seconds in our case). However, the transport solver has a time step of 1 minute, therefore, moving the emission and dry deposition from the chemical solver to the transport solver reduces the run time for the gas-phase chemistry by 70%, making the model more computationally efficient for adding increasingly complex chemical mechanisms in the future. In summary, the differences of the modeled gradients in isoprene concentrations between the two operator orders are relatively small (up to 16%) with the new order having a more realistic in-canopy profile and a higher

computational efficiency.

The impacts of the operator splitting on the gas concentrations are correlated with the chemical lifetime (Fig. 2). The OH and nitric oxide (NO) concentrations increase by 160-180% and 130% with the new order of operator, respectively, while CO (carbon monoxide) only differs by 13% (Fig. 2). The increased NO, $CH_2O$ (formaldehyde), and $O_3$ using the new order improve agreement with the observations (Fig. 4). In addition, the vertical gradients in concentrations between in- and above-

canopy decrease with increasing lifetime with 20% difference for OH and almost zero for CO (Fig. 2). Overall, the results here draw attention to the influences of the operator splitting on reactive trace gases such as OH and NO, which are critically important for accurately predicting the gas-phase chemistry and aerosol formation. In summary, the advantages of the new operator order from the modeling perspective includes (i) generally better agreement with the observations for the critical species NO and OH, (ii) improved in-canopy gradients for emitted species such as isoprene, and (iii) higher computational

efficiency. Therefore, we implement the new order of operations in FORCAsT 2.0.

## 3.2   Revised vertical mixing

The revised vertical mixing parameterization produces a larger eddy diffusivity (K) and a more realistic boundary layer height (Fig. 3a). The parameterization of K in the boundary layer remains challenging, as K is a derived parameter that is analogous to diffusion yet is not completely accurate for boundary layer turbulent mixing. Despite its limitations, it is still a useful

approximation when a full solution for turbulence is computationally expensive to implement. Kumar and Sharan (2012) compiled estimated values for K in the boundary layer based on previous studies, suggesting the magnitude of the K ranges from 60 to 200 m$^2$s$^{-1}$ under weakly unstable conditions (z/L=-2). In FORCAsT 1.0, K peaks at 16 m$^2$s$^{-1}$ at 150 m above



the ground during the daytime (Fig. 3a). This leads to weak mixing, resulting in an unrealistic end-of-day peak in isoprene concentrations around sunset for the well-mixed day of July 22 (Fig. 3c). The revised parameterization produces a larger K

that falls within the lower range of previously reported values (Kumar and Sharan, 2012) and is sufficiently strong to produce a realistic isoprene diurnal cycle under well-mixed conditions (Fig. 3c). On the stagnant second day of the simulation (July 23), the revised parameterization also reproduces the end-of-day peak in isoprene but the modeled isoprene is much lower than the observed isoprene (Fig. 3c).

FORCAsT predicts the air temperature based on mixing of surface heat instead of prescribing the measured temperature

to nudge the model; therefore the vertical mixing impacts the temperature profiles (Fig. 3b). During the daytime the canopy behaves as a heat source based on the leaf energy balance, and the stronger mixing distributes the heat more evenly throughout the model atmosphere. Therefore, a larger K reduces air temperature during the daytime in FORCAsT (Fig. 3b). At night the canopy is cooler than the air aloft due to the longwave radiation emission from the canopy and the low nocturnal mixing ($K < 3.5$ $m^2 s^{-1}$, not shown) fails to mix the warmer air down to the canopy, resulting in nighttime cooling of $8^{\circ}C$ during

0-6 model hours and a smaller minimum temperature than the observation around 30 model hours (Fig. 3b). The unrealistic nighttime cooling during 0-6 model hours indicates that (i) heat capacity of leaves may be important as a heat source at night; and/or (ii) nighttime mixing is too low in FORCAsT. The idea of low nighttime mixing is also supported by the overestimated nighttime isoprene concentrations (Fig. 3c), suggesting the need for a better data-constrained nighttime mixing scheme. Note that air temperature (Thomas, 2011) and evening isoprene decay (Wei et al., 2020) also depend on horizontal advection which

is not considered in this study. Attention should be on these estimates when the homogeneity assumption is not met at the site, particularly under stable conditions.

### 3.3 Isoprene updates to the gas-phase chemistry mechanism

RCIM incorporates the current knowledge of isoprene chemistry under low-NO conditions, which is most notably manifested by the changes in the oxidation products of isoprene (Fig. 4). Adding the RCIM isoprene chemistry does not drive large

changes in simulated isoprene concentrations (Fig. 4a), which respond to OH, the isoprene emission rate, and mixing, yet there is a substantial increase in MVK+MACR (Fig. 4c). Generally the RCIM mechanism simulates MVK+MACR that shows better agreement with the observations, although simulated concentrations are lower than observed on the second day which is likely due to underestimated isoprene (Fig. 4a). The measured MVK+MACR peaks before isoprene on the first day, suggesting horizontal advection and/or vertical transport from the residual layer. Both RCIM and CACM1.0 underestimate formalde-

hyde ($CH_2O$), a high-yield product of isoprene oxidation, by over 50% (Fig. 4j). Marvin et al. (2017) show existing isoprene mechanisms generally underestimated $CH_2O$ by $17-33\%$. This suggests missing sources for $CH_2O$, possibly heterogeneous conversion of ISOPOOH on leaves and/or unaccounted for VOC chemical oxidation in the chemical mechanisms (Canaval et al., 2020; DiGangi et al., 2011).

The modeled NO, $NO_2$ and $O_3$ are very similar between the two mechanisms (Fig. 4d-f). The timing of the early morning

NO maxima, caused by photolysis of $NO_2$ transported downward during the morning breakup of the nocturnal boundary





layer (Seok et al., 2013), are captured by both mechanisms. Both mechanisms overestimate $NO_2$ during the 0-6 model hours, suggesting missing sinks for $NO_2$, possibly the aqueous phase reaction of $NO_2 + NO_3 \rightarrow N_2O_5$.

Adding RCIM also impacts $HO_x$, with little change in $HO_2$ (Fig. 4h) but a substantial increase in OH (by 50% on the second day, Fig. 4g). Both mechanisms have HOx-regeneration from the H-shift isomerization of isoprene peroxy radicals (ISOPOO),

which is important to sustain the OH concentrations under low-NO conditions (Peeters et al., 2009; Bates and Jacob, 2019). CACM1.0 recycles 1.0 $HO_2$ via the 1,6-H shift, while RCIM recycles 1.5 OH + 0.7 $HO_2$ through the 1,6-H shift pathway and 1.0 OH through the 1,5-H shift pathway (3.2 equivalents to $HO_x$). Note that the 1,6-H shift pathway dominates the 1,5-H shift pathway by a factor of 8 in RCIM. In addition, the ISOPOO concentrations in CACM1.0 are lower than in RCIM by 50% (Fig. 5a). Therefore, the combination of higher ISOPOO and OH-regeneration efficiency leads to larger OH concentrations in

RCIM. Overall, the modeled OH is in good agreement with the campaign-average measurements (Figure 4g).

RCIM predicts higher daytime $RO_x$ ($HO_2 + RO_2$) than CACM1.0 (Fig. 4i), which is predominantly due to differences in ISOPOO (Fig. 5a). ISOPOO accounts for 50% and 25% of $RO_x$ in RCIM and CACM1.0, respectively. In the early morning hours (0−8 local time) $RO_x$ derive from $NO_3$-initiated reactions with monoterpenes (58% from $\alpha$-pinene and $d$-limonene, and the remaining $RO_x$ from oxidation products of $d$-limonene such as limonaldehyde and limona ketone). The observed $NO_3$ is

below the limit of detection of the instrument (1.4 pptv) on the two simulated days (not shown), but both mechanisms simulate $NO_3$ in the range of 6−7.5 pptv for this period (0−8 model hours) in response to the high simulated $NO_2$ and $O_3$ (Fig. 4e,f). Simulated $NO_3$ decreases and is comparable to the limit of detection during model hours 19−30, but the corresponding $RO_2$ is still overestimated with 40% ISOPOO and 60% monoterpene-derived $RO_2$, suggesting missing sinks for nighttime $RO_2$. One potential sink is reaction with NO. Using the observed NO to constrain the model reduces the nighttime $RO_x$ by roughly

30% (not shown), but this is not sufficient to reproduce the observed $RO_x$. Another possible sink is the accretion reactions of monoterpene-derived $RO_2$. The most recent accretion reaction rates for OH-initiated and $O_3$-initiated monoterpene $RO_2$ are included in FORCAsT ($3.7 \times 10^{-11}$ and $9.7 \times 10^{-12}$ $cm^3$ molecule$^{-1}$ s$^{-1}$, respectively; Berndt et al., 2018). However, laboratory updates on accretion rates for $NO_3$-initiated monoterpene $RO_2$ that dominate at night are not available. Finally, dry deposition for $RO_2$ is not included in the model due to lack of data, and this has the potential to be an additional nighttime sink.

Overall, the results suggest that a better understanding of nighttime sinks of $RO_x$ is needed, including chemical losses and dry deposition.

Because of the numerous mechanism changes for low-NOx isoprene chemistry, large deviations between RCIM and CACM1.0 occur for the isoprene oxidation products, including ISOPOOH (isoprene hydroxy hydroperoxide), IEPOX (isoprene epoxydiol), and IHN (isoprene hydroxy nitrate) (Fig. 4k−n). ISOPOOH, formed by the reaction of $HO_2$ with ISOPOO, is increased

by a factor of 2 in RCIM throughout most of the diurnal cycle (Fig. 4k) due to a combination of higher ISOPOO concentrations and higher fraction of ISOPOO following the $HO_2$ pathway (Fig. 5). IEPOX, predominantly produced by the reaction of ISOPOOH + OH, is slightly higher in RCIM (Fig. 4l).

One important difference between the two mechanisms is the ISOPOO-NO reaction pathway increases in RCIM (29%) as compared to CACM1.0 (11%) (Fig. 5b,c). From a low-NO study in the Amazon (NO < 0.1 ppbv), Liu et al. (2016) found

that the ratio of $HO_2$ pathway to NO pathway is about unity. For these UMBS simulations with low NO conditions, RCIM





shows a similar NO:HO$_2$ ratio of 1.3, in contrast to the ratio of 3 in CACM1.0 (Fig. 5b,c). The higher NO pathway percentage in RCIM consequently increases IHN concentrations (Fig. 4 m, n). The in-canopy IHN (19.5 m) measured by iodide-adduct chemical ionization mass spectrometer (CIMS) is well reproduced by RCIM, while the above-canopy IHN (32 m) by GC-HR-ToF-CIMS is overestimated by a factor of 2. Because we do not simulate strong vertical gradients in isoprene (supported by

observations, e.g., Wei et al., 2020), the IHN differences in the observations are not likely due to vertical mixing or horizontal advection. Vasquez et al. (2020) show the 1,2-IHN isomer undergoes a rapid hydrolysis loss not experienced by the 4,3-IHN isomer. The in-canopy (19.5 m) measurements by the iodide-adduct CIMS may be subject to uncertainty in the hydrolysis related isomer sensitivity (Fig. 4m,n). Vasquez et al. (2020) estimate a condensed-phase hydrolysis coefficient of $4\times10^5$ M atm$^{-1}$ s$^{-1}$ for 1,2-IHN to match their observed 1,2-IHN to 4,3-IHN isomer ratio. Using this hydrolysis loss coefficient, our

model is able to reproduce the isomer ratio. However, the resulting total IHN (1,2-IHN+4,3-IHN) are still significantly higher than the measurements (dashed line in Fig. 4m, n). A hydrolysis coefficient of $1\times10^8$ M atm$^{-1}$ s$^{-1}$ is required for the model to match the measured 1,2-IHN concentrations at 32 m, however, the ratio is then underestimated because the model simulates more 4,3-IHN than observed(not shown), which may be due to the greater production of the 4,3-IHN driven by higher than observed HO$_2$. Overall, the differences in IHN observations as well as the measured-modeled discrepencies in the IHN isomers

suggest that large uncertainties still exist in the simulated IHN concentrations.

The monoterpene hydroxy nitrates (MTN) in RCIM are slightly lower than that in CACM1.0 at night (Fig. 4o), likely driven by lower nighttime monoterpene concentrations in RCIM (Fig. 4b). Overall, the MTN is underestimated by both RCIM and CACM1.0, and this could be affected by the rapid monoterpene-OH RO$_2$ accretion reactions, which decrease the available monoterpene RO$_2$ for generating MTN during the daytime. Additionally, the dry deposition of MTN is prescribed to be 50%

of IHN, which lack of observational constraints and may result in an over-consumption of MTN. To further evaluate the representation of the low-NO chemistry of isoprene by the two mechanisms, we compare the ratio of ISOPOOH to MVK+MACR. For unpolluted regions, the reaction of ISOPOO with HO$_2$ is the dominant pathway with ISOPOOH isomers as the major oxidation products. Reaction of ISOPOO with NO dominates in polluted regions, with major oxidation products MVK and MACR. MVK+MACR can also be produced through the isomerization and self-reaction of ISOPOO, with MVK dominating

over MACR by approximately a factor of 2. Therefore, the ratio of ISOPOOH to MVK+MACR reflects the contribution of the HO$_2$ oxidation pathway relative to other oxidation pathways of ISOPOO. The consistency between the measured and modelled ISOPOOH fraction thus reflects the skill of the isoprene mechanisms in representing the isoprene chemistry under low-NO conditions. The most frequent (51 %) observed daytime NO levels when the ISOPOOH data are available ranges from 20 to 40 pptv. The measurement-based ratio of ISOPOOH to MVK+MACR decreases quickly ranges from 0.15 to 0.05 with increasing

NO when NO<40 pptv (Fig. 6), indicating the fate of ISOPOO is highly sensitive to NO levels under low-NO conditions. The simulated dependence of the ratio on NO concentrations by RCIM is in good agreement with the observed, although the simulated ratio decays faster than the observations. In contrast, the ratios simulated by CACM1.0 are higher and decrease slower with increasing NO concentrations, partly due to lower OH (Fig. 6) that reacts faster with ISOPOOH than with MVK.

In summary, the fate of ISOPOO in RCIM is different than in CACM1.0 (Fig. 5), leading to differences in isoprene oxida-

tion products. The HO$_2$ pathway ratio is higher in RCIM, leading to higher ISOPOOH concentrations, and the NO pathway of





ISOPOO is also enhanced in RCIM, resulting in higher MVK+MACR and IHN concentrations. This comes at the expense of reducing the H-shift isomerization pathway in RCIM, yet OH concentrations are still enhanced due to higher $HO_x$-recycling efficiency in RCIM. Overall, isoprene oxidation products in RCIM generally compare well with observations and RCIM captures the decreasing trend of ISOPOOH/(MVK+MACR) with increasing NO concentrations, indicating a better representation

of low-NO chemistry. One aspect to note about the one-dimensional framework is that it prioritizes local processes (emissions, chemistry and deposition) over that of advection. If we assume a horizontally homogeneous canopy as utilized in flux analysis, then advection should not be a factor. However, observations are clear that the site can be influenced by long-range transport, such as the advection of high NOx conditions from southern urban locations such as Detroit, Milwaukee, and Chicago (e.g., Cooper et al., 2001; VanReken et al., 2015). One factor to consider is if horizontal advection also influences the diurnal

cycle of isoprene and monoterpene oxidation products. For example, above-canopy short-lived oxidation products (such as ISOPOOH and IHN) are overestimated in the model compared the observations, whereas longer-lived species such as IEPOX (Fig. 4k−m) show better agreement with the observations. Further analysis is needed that includes upwind sources of these oxidation products, as well as potential analysis with wind direction and VOC sources.

### 3.4 Secondary organic aerosol formation

As discussed in Section 2.5, an isoprene-derived SOA (iSOA) parameterization based on Griffin et al. (2005) is implemented in FORCAsT 2.0, incorporating 6 new surrogate groups for the isoprene-derived precursors such as IEPOX (isoprene epoxydiol) and tetrafunctionals. As described in Ashworth et al. (2015), the partitioning of organic gases into the aerosol phase is a function of the liquid water content (Equation 11).

   Both monoterpene-SOA (MNT-SOA) and iSOA mainly form in the boundary layer (∼1.5 km; Fig. 7a, b) in response to ver-

tical mixing, vertical distributions of SOA precursors, and meteorological conditions such as relative humidity. The simulated boundary layer-averaged MNT-SOA is roughly 0.15±0.1 $\mu$g m$^{-3}$. The MNT-SOA above the canopy (30 m) is underestimated by a factor of 2-3 compared to the HR-ToF-AMS positive matrix factorization concentrations (Fig. 3c), which is likely due in part to the underestimated precursors such as MTN (Fig. 4o). Another possible reason is the partitioning efficiency for MNT-SOA is underestimated due to phase separation in MPMPO. MPMO considers the two phases separately: the organic

phase and the aqueous phase. In reality, water uptake to organics increases the partitioning medium, which in turn increases the partitioning efficiency of condensable organic species (Pye et al., 2017), and this could effectively reduce the simulated MNT-SOA in MPMPO.

   The majority of simulated iSOA (>99%) is formed through the aqueous phase, and therefore is sensitive to the liquid water content (Fig. 7b,e,f). The boundary layer-averaged iSOA is 1.0±1.0 $\mu$g m$^{-3}$ (Fig. 7b). IEPOX-SOA dominates the modeled

iSOA formation (Fig. 8a,b), and simulated concentrations agree well with the daytime HR-ToF-AMS PMF-derived IEPOX-SOA at 30 m (Fig. 7d). The modeled IEPOX-SOA show a diurnal cycle that follows gas-phase IEPOX concentrations (Fig. 4l), and the rapid increase in IEPOX-SOA around 35 model hours result from the increase in liquid water content (Fig. 7e). No such increase in IEPOX-SOA is manifested around 6 model hour when liquid water content is also high due to absence of iSOA precursors at the beginning of the simulation (Fig. 4l).





FORCAsT 2.0 simulates an in-canopy iSOA mass yield of 7%, which is slightly higher than values commonly used in global models (0.9 %−6.8 %) to represent the ambient atmosphere (Carlton et al., 2009). In the model, the two dominant iSOA surrogate groups are the tetrafunctionals (Tetra-SOA) and IEPOX (IEPOX-SOA), which combined account for 86% of the total iSOA in the canopy and 98% in the boundary layer (Fig. 8a, b). RCIM has a a low yield of glyoxal from isoprene and the GLYX-SOA component is 4 % in the canopy and negligible in the boundary layer. The iSOA from organic nitrates (IHN-SOA

and C4-SOA) are more important (11%) in the canopy likely due to higher NO levels (Fig. 8a, b). Both IEPOX-SOA and Tetra-SOA increase with height (Fig. 8c,d) following the gas-phase precursors (not shown). We note that dry deposition of tetrafunctionals lacks data constraints, and in this study, the dry deposition of these compounds are set to be the same as IHN (Table 2) due to the common nitrate functional group. This leads to the deposition of 60% of the gas-phase tetrafunctionals in the canopy and is a major source of uncertainties for estimation of Tetra-SOA. Using RCIM, Bates and Jacob (2019) report an

even higher iSOA mass yield of 25 % with IEPOX-SOA, organonitrates, and Tetra-SOA each contributing about one-third to the total mass. The differences in the iSOA yield and composition between this study and Bates and Jacob (2019) may derive from (i) the $NO_x$ levels which are expected to significantly impact the iSOA precursors (i.e. gas-phase IEPOX, tetrafunctionals, and organic nitrates); or/and (ii) the different iSOA formation schemes.

    There are a few points to note in our estimation for LWC. The $LWC_{org}$ accounts for almost 40% of total LWC on average

(Fig. 7e), suggesting water uptake to organics are important for iSOA formation at the site. However, $LWC_{org}$ is highly sensitive to the hygroscopy growth for organics ($\kappa_{org}$) ranging from 0.01 to 0.5 for slightly to very hygroscopic organic species (Petters and Kreidenweis, 2007). Our calculation gives a mean $\kappa_{org}$ of 0.08 for the simulation period. Cerully et al. (2015) partitioned $\kappa_{org}$ for different organic aerosol compositions with 0.08±0.02 for monoterpene-SOA and 0.20±0.02 for isoprene-SOA, respectively. To test the impact of $\kappa_{org}$ on LWC estimation, we run a sensitivity simulation with a higher $\kappa_{org}$ of 0.15, resulting

in an increase of 0.4 $\mu$g m$^{-3}$ in $LWC_{org}$ on average and thus a relatively small increase in total LWC (Fig. 7e). Therefore, the higher $\kappa_{org}$ of 0.15 leads to a negligible increase on iSOA formation. Slade et al. (2019) derived different $\kappa_{org}$ values for daytime (0.02) and nighttime (0.15) due to diurnal changes in organic aerosol composition at the same study site, which would require a diurnal cycle of $LWC_{org}$ with lower values at night. Because this result contrasts with current methods to derive LWC, we were unable to test whether this would impact our partitioning simulations, but note that this large discrepancy could

have implications for the simulation of biogenically-derived organic aerosol over the diurnal cycle. Finally, we note that due to lack of measurements outside of the canopy, the vertical distribution of LWC is scaled by relative humidity (RH) in the model. During the daytime (nighttime), LWC is relatively constant within the boundary (surface) layer and decreases rapidly with altitude above the boundary (surface) layer (Fig. 7f). In general, this trend is consistent with the meteorology soundings of RH from the nearest location (in Gaylord, Michigan), but the soundings suggest the daytime boundary layer height and the

nighttime surface layer height are underestimated in the model despite the improved vertical mixing formulation.





## 4  Conclusions

We update FORCAsT 1.0 to FORCAsT 2.0 by including (i) splitting the integration of chemistry from emission and dry deposition to provide more realistic representations of vertical gradients in the forest canopy and to make the chemical module more flexible for future chemical mechanism changes, (ii) updating eddy diffusivity in the boundary layer and the dry deposition

velocity with available measurements; (iii) implementing the Reduced Caltech isoprene mechanism (RCIM) that reflects the current understanding of isoprene fate under low $NO_x$ conditions; (iv) extending the aerosol module to include isoprene-derived SOA (iSOA) formation.

The new order of operations reduces the run time for the gas-phase chemistry by 70%, making the model more computationally efficient for adding increasingly complex chemical mechanisms in future. Meanwhile, the differences of the modeled

gradients in isoprene concentrations between the two orders are relatively small (up to 16%) with the new operator order having a more realistic in-canopy profile. The revised mixing parameterization yields greater eddy diffusivity in the boundary layer and improves the estimation of the diurnal cycles of isoprene. The end-of-day peak in isoprene is removed for the well-mixed conditions and retained for the stagnant conditions. The revised mixing also improves the simulated daytime air temperature. The revised dry deposition improves the deposition velocity for species when measurements are available.

The fate of isoprene peroxy radicals (ISOPOO) in RCIM is different than in CACM1.0. The key differences include a higher fraction of ISOPOO following the NO pathway (29%), leading to higher concentrations of MVK+MACR (by a factor of 2) and in-canopy IHN (by a factor of 8) in RCIM. Despite a lower ratio of H-shift isomerization pathway (26%), RCIM recycles more $HO_x$ than CACM1.0. In addition, the ratio of ISOPOOH to MVK+MACR, which reflects the skill of the mechanisms in representing the isoprene chemistry under low-NO conditions, decreases from 0.15 to 0.02 for NO levels from 20 to 240 pptv.

The decreasing trend of the ratio with increasing NO is generally reproduced by RCIM, indicating an improved representation of the chemistry under low-NO condition. FORCAsT 2.0 suggests a 7 % mass yield of isoprene-derived secondary organic aerosol (iSOA) in the canopy. The tetrafunctional-derived iSOA (Tetra-SOA) and IEPOX-derived iSOA (IEPOX-SOA) combined account for >86% of the total iSOA, with IEPOX-SOA dominant by a factor of 3.5. The iSOA from organic nitrates are more important in the canopy than above the canopy, accounting for 11% of the total iSOA.

Generally, FORCAsT 2.0 shows better agreements with the observations. However, several limitations remain. Due to challenges in nighttime turbulence measurements, the poor simulation of nighttime mixing likely contributes to the cool bias in nighttime air temperature and overestimated isoprene concentrations. While there are many improvements in the evaluation of isoprene chemistry, there are still some notable discrepencies between the observations of isoprene oxidation products and both chemical mechanisms, including first generation products such as ISOPOOH and $CH_2O$, as well as second generation prod-

ucts such as IEPOX. Additionally, uncertainties in the biogenically-derived nitrates (IHN and MTN) suggest that additional observations are necessary to understand the pathways of formation and removal. Heterogeneous reactions are not included in the current version, which could otherwise improve the simulation of nighttime $NO_3$, $CH_2O$, and IHN. For the formation of biogenically-derived aerosols, some thermodynamic parameters (such as UNIFAC parameters and Henry's Law constants) for the new surrogate groups (such as nitrate and hydroperoxide) carry uncertainties and better constraints from laboratory data



could improve the simulation of iSOA yields. The phase separation in MPMPO also adds uncertainty to SOA estimation. The diurnal cycle and vertical distribution of liquid water content lack data support and therefore represent additional sources of uncertainty for SOA estimation. Finally, dry deposition for the tetrafunctional compounds and $RO_x$ are poorly constrained due to lack of information. More generally, this study assumes horizontal homogeneity at the study site, which might not be true due to the heterogeneity of the land cover caused by surrounding geography which includes several large freshwater lakes.

This problem is particularly critical under stable conditions. Further analysis of the role of horizontal advection at the site may be useful in understanding how long-range transport influences the local concentrations of long-lived biogenic VOC oxidation products. Taken together, these limitations carry certain levels of uncertainties which can be addressed in future development. Despite these limitations and uncertainties, the revised 1-D model provides a useful tool for understanding how local conditions can influence the oxidation of biogenic VOC and subsequent aerosol processes.

*Code availability.* https://github.com/steiner-lab/FORCAsTv2

*Data availability.* Isoprene, monoterpenes, and MVK+MACR are available online at https://doi. org/10.13020/D6JQ3R (Alwe et al., 2019). The GC-HRToF-CIMS ISOPOOH, IEPOX, and IHN are available through Vasquez et al. (2018) (https://data.caltech.edu/records/971).

*Author contributions.* ALS designed the study. DW implemented the model updates and performed the simulations and analysis. HDA and DBM provided the VOCs and turbulence data. BB, ML, and PSS provide the OH and HO2 data. JDS, JLC, FNK provided the HCHO data.
QS, SCK, and JGM provided the $NO_x$ data. KTV, HMA, EP, JDC, and POW provided above-canopy ISOPOOH and IHN data. JDC and POW provided critical review on model evaluation regarding IHN. PBS provided the in-canopy IHN and MTN data and critical review on the inclusion of liquid water content associated with organics. AATB, HWW, and RJG provided the AMS data. RJG provided critical review on the MPMPO module. NWM, MC, JHS, and KAP provided the AIM data. KAP contributed to the calculation of liquid water content associated with inorganic species. ECW, MR, BLD, and DCA provided the $RO_x$ data. All authors contributed to model evaluation and the
manuscript preparation.

*Competing interests.* The authors declare that they have no conflict of interests.

*Acknowledgements.* DDW and ALS acknowledge support by National Oceanic and Atmospheric Administration under grant NA18OAR4310116. Andrew P. Ault and Ryan Cook (University of Michigan) are acknowledged for field campaign assistance. K.A. Pratt, N.W. May, and M. Conner acknowledge support from the University of Michigan MCubed 2.0 program and the Marian P. and David M. Gates Graduate Student
Endowment Fund. The construction of the GC-HRToF-CIMS instrument was supported by the National Science Foundation (AGS-1428482),



with additional NSF support (AGS-1240604) provided for the instrument field deployment to UMBS. Work performed by Krystal T. Vasquez and Hannah M. Allen was supported by the National Science Foundation Graduate Research Fellowship (NSF GRFP). Krystal T. Vasquez also acknowledges support from an Earl C. Anthony Fellowship in chemistry during an early portion of this study. The measurements of peroxy radicals were supported by NSF AGS-1443842 to University of Massachusetts Amherst and AGS-1719918 to Drexel University.

Work performed by Joshua D. Shutter and Joshua L. Cox was supported by the National Science Foundation Graduate Research Fellowship (NSF GRFP). FNK acknowledges support by the National Science Foundation (AGS-1643306).



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



**Figure 1.** Impacts of operator-splitting on the vertical distribution of isoprene. The modeled and observed isoprene vertical profiles (normalized by the concentration at the canopy) in the morning **(a)** , mid-day **(b)**, and evening **(c)** on July 22, 2016. Note that the observed isoprene at 34, 21, 17, 13, 9, and 5 m are interpolated into model levels for comparison. The corresponding Root Mean Square Error (RMSE) of the modeled isoprene profiles compared to the observations for the morning **(d)** , mid-day **(e)**, and evening **(f)** on July 22, 2016. The RMSE are also calculated separately for the in-canopy (**In-cpy**) and above-canopy (**Abv. cpy**).



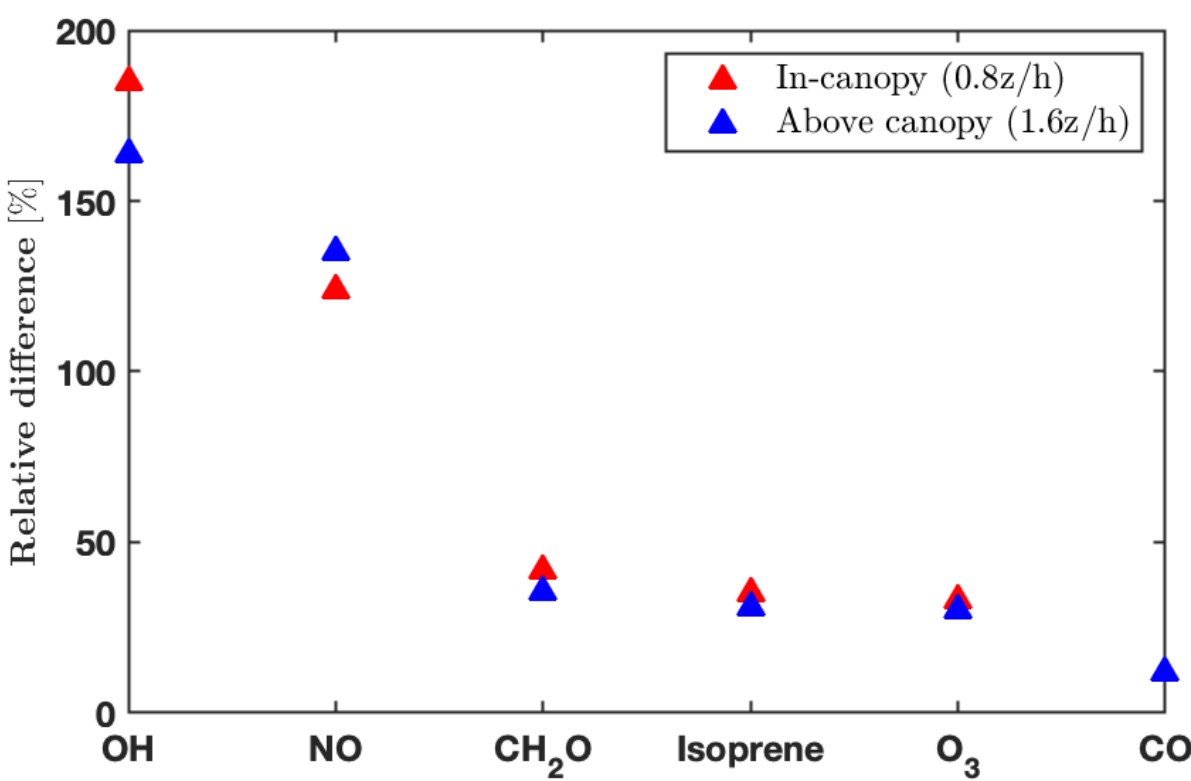

**Figure 2.** Impacts of operator-splitting on species with chemical lifetimes ranging from seconds (OH, Di Carlo et al., 2004) to months (CO, Holloway et al., 2000). The relative differences between the two orders of operator (i.e. $\frac{C_{new} - C_{old}}{C_{old}}$) for daytime (10:00 am - 4:00 pm local time) average hydroxyl radicals (OH), nitric oxide (NO), formaldehyde (CH$_2$O), isoprene, ozone (O$_3$), and carbon monoxide (CO) at two heights (0.8 and 1.6 z/h).



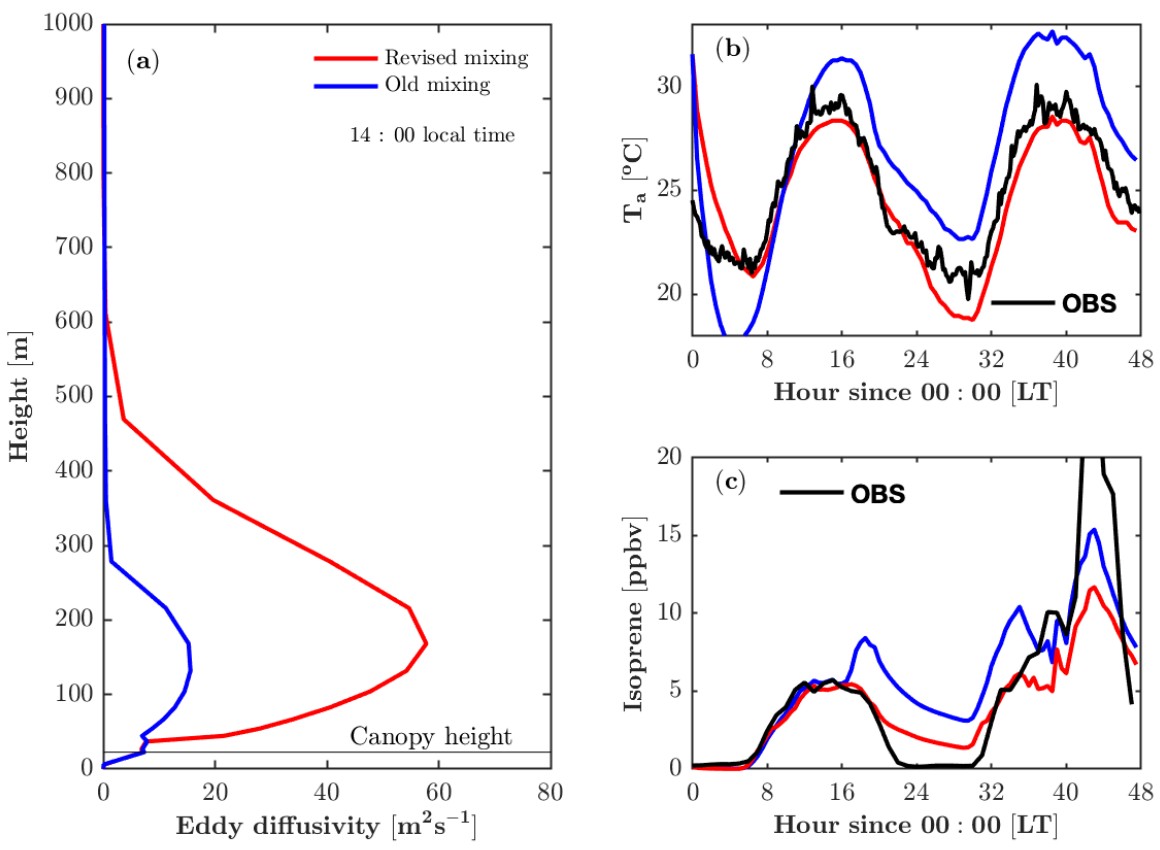

**Figure 3.** Impacts of the revised mixing parameterization on air temperature and isoprene. **(a)** Modeled vertical profile of the eddy diffusivity at 14:00 local time on July 22 2016. **(b)** Modeled and observed air temperature ($T_a$) at 46 m. **(c)** Modeled and observed isoprene at 21 m.





**Figure 4.** Comparison of FORCAsT 2.0 with the RCIM (**red**) and CACM1.0 (**blue**) mechanisms against observations in and above the canopy (heights noted). Gas-phase concentrations are evaluated versus observations (**grey**), including isoprene (34 m), monoterpenes (34 m), methyl vinyl ketone + methacrolein (MVK+MACR, 34 m), nitric oxide (NO, 29 m), nitrogen dioxide ($NO_2$, 29 m), ozone ($O_3$, 6 m), hydroxyl radicals (OH, 32 m), hydroperoxy radicals ($HO_2$, 32 m), total peroxy radicals ($RO_x$, 30 m), formaldehyde ($CH_2O$, 21 m), hydroxy hydroperoxide isomers (1,2-ISOPOOH + 4,3-ISOPOOH, 32 m), epoxydiol isomers (cis-IEPOX+rans-IEPOX, 32 m), isoprene hydroxy nitrates (1,2-IHN + 4,3-IHN, 32 m), IHN (19.5 m), monoterpene hydroxy nitrates (MTN, 19.5 m)in sequence from **(a)** to **(o)**. Grey shaded areas denote one standard deviation of the data when available. Instrumental information is in Table 4. The red dashed lines in (m) and (n) denote modeled IHN with a hydrolysis loss rate of $4\times10^5$ M $atm^{-1}$ $s^{-1}$ for 1,2-IHN.



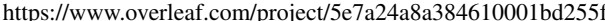

**Figure 5.** Fate of isoprene hydroxy peroxy radicals (ISOPOO) in RCIM and in CACM1.0. **(a)** Diurnal profiles of ISOPOO concentrations in RCIM and CACM1.0. **(b, c)** Fraction of daytime-averaged (12:00-14:00 LT) production rates of ISOPOO through reactions with peroxy radicals (RO$_2$), hydroperoxy radicals (HO$_2$), nitric oxide (NO), and H-shift isomerization in RCIM and CACM1.0.

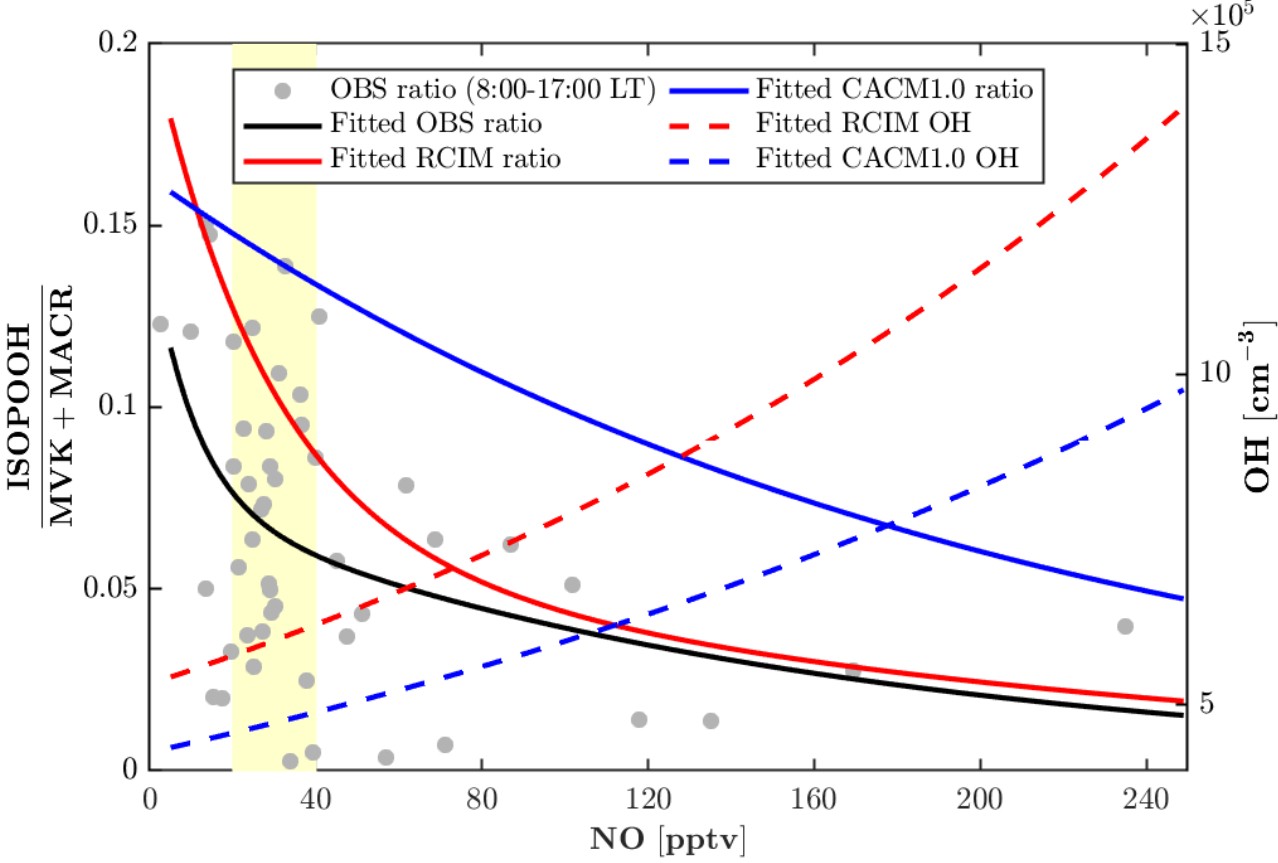

**Figure 6.** Daytime (8:00-17:00 local time) measurement-based and modeled ratio of ISOPOOH (isoprene hydroxy hydroperoxides) to MVK+MACR (methyl vinyl ketone + methacrolein). Measurement-based ratio of ISOPOOH to MVK+MACR are from 23−27 July 2016 (**Grey dots**). Fitted exponential curve for measurement-based ratios (**black line**), RCIM ratios (**red line**), CACM1.0 ratios (**blue line**), RCIM OH concentrations (**red dash line**), and CACM1.0 OH concentrations (**blue dash line**). Yellow patch denotes the most frequently observed nitric oxide (NO) levels (20−40 pptv) during the period of 23−27 July 2016.





**Figure 7.** Monoterpene-derived secondary organic aerosols (MNT-SOA) **(a)** and isoprene-derived secondary organic aerosols (iSOA) **(b)** as a function of time and height. **(c, d)** Comparison of modeled MNT-SOA and IEPOX-SOA with observations at 30 m. **(e)** Time series of calculated liquid water content (LWC) with different hygroscopicity growth $\kappa$. **(f)** Vertical profiles of modeled LWC during the daytime (14:00 local time) and at early morning (6:00 local time) .



**Figure 8.** Composition of the average isoprene-derived secondary organic aerosols (iSOA) over the two-day period at two different heights. The composition components are the surrogate groups of GLYX (glyoxal and methylglyoxal), IEPOX (isoprene-epoxydiol, isoprene-carbonyl-hydroxy-epoxide, and hydroxy methyl methyl-a-lactone), IHN (isoprene hydroxy nitrates), INP (nitrooxy hydroperoxides), C4 nitrate, and tetrafunctionals. Details about the surrogate groups can be found in Section 2.5. Composition of iSOA at 19 m **(a)** and 500 m **(b)**. Isoprene-epoxydiol aerosols (IEPOX-SOA, **c**) and tetrafunctional compound aerosols (Tetra-SOA, **d**) as a function of time and height.