# Peer review of "FORest Canopy Atmosphere Transfer (FORCAsT) 2.0: model updates and evaluation with observations at a mixed forest site"

_Geoscientific Model Development, 2021_

## Author Response (AR1)

**Responses to the executive editor**

We moved the codes to Zenodo (https://doi.org/10.5281/zenodo.4776662), and have added a GPLv3 license as recommended. The codes are now citable with a DOI of 10.5281/zenodo.4776662. The 'Code and Data availability' section is revised to include the new Zenodo repository reference.

**Responses to Reviewer#1**

General comments
This study updated the 1D model FORCAsT from 1.0 to 2.0 by improving the computation efficiency, eddy diffusivity parametrization, dry deposition calculation, as well as the isoprene-related chemistry and aerosol processes. The comparison results with the measurement data showed that the new model performed better especially in simulating, e.g., the vertical profile of isoprene concentration in the early morning, diurnal variation of temperature, the in-canopy concentrations of MVK+MACR and IHN, the ratio of ISOPOOH to MVK+MACR under low-NO condition. This research fits the scope of GMD, and the manuscript is written clearly, so I suggest to publish after minor revision.

Specific comments
P6, L137: So here z_i is an integral value, is it a constant within one day? How do you calculate K_new before z_i is calculated every day? Please specify more clearly here how the K_new and z_i are calculated in the model.

The boundary layer height ($z_i$) is not constant during the daytime, and it varies as a function of the above canopy heat flux $(w'\theta')_{cpy}$ (see Equation 8). The variable $z_i$ is calculated before K_new at every time step. The integral $\int_0^t (w'\theta')_{cpy}$ is approximated to $(w'\theta')_{cpy} \cdot \Delta t$, where $\Delta t$ is the elapsed time. The boundary layer height z_i is calculated only in the when $(w'\theta')_{cpy}$ is positive. For example, if the $(w'\theta')_{cpy}$ becomes positive at 6:00 am as the sun rises, the $\Delta t$ for calculation of $z_i$ at 7:00 AM is 1 hour=3600 s. Canopy heat fluxes tend to be negative at night, and therefore the nighttime $z_i$ is set to 200 m.

In response to this comment, we revised the text to clarify the calculation of the integral (lines 138-141):

> "The integral $\int_0^t (w'\theta')_{cpy}$ is approximated to $(w'\theta')_{cpy} \cdot \Delta t$, where $\Delta t$ is the elapsed time. The calculation for $z_i$ starts when the heat flux $(w'\theta')_{cpy}$ first becomes positive in the morning and continues one more hour after $(w'\theta')_{cpy}$ becomes negative. The nighttime $z_i$ is set to 200 m."

P7, L152: "where input data are available".
What are the input data for the calculation of dry deposition velocities of the species here?
P7, L158: "lack of data". What kind of data does it mean?

The "input data" for the revised resistance model are shown in Table R1.

Table R1. Input data for calculation of the three resistances: aerodynamic resistance (Ra), molecular diffusion resistance (Rb), and surface resistance (Rc). $\kappa$ is the von Kármán constant (0.4), $u_*$ is the friction velocity from the measurements, U is the wind speed from the measurements, $\nu$ is the viscosity of air, $D_x$

is the diffusivity of a molecule X in air, l is taken to be 0.001, LAI is the leaf area index, $r_{H2O}$ is the stomatal resistance to the diffusion of water, H is the Henry's Law coefficient, f0 is a reactivity factor.

| Variables | Input data |
|---|---|
| Ra | $\kappa$, $u_*$, U, z/L |
| Rb | $\nu$, $D_x$, $u_*$, l, LAI |
| Rc | $D_x$, $D_{H2O}$, $r_{H2O}$, H, f0, R, T |

Because this model parameterization is published elsewhere, we have not included this table in the main manuscript but do clarify the input data here for the reviewer.

Among this list of input variables, the resistance model is highly sensitive to Henry's Law coefficient (H). Uncertainties in H can be >100% due to the paucity of measurements (Fig. S16 from Nguyen et al. 2015, see below). Therefore, the meaning of "where input data are available" is two-fold. First, a reliable Henry's Law coefficient (H) is needed. Second, direct estimates of dry deposition velocity (Vd) from measurements (through the ratios of fluxes to concentrations) are needed to validate the revised resistance model. For species whose H and measurement based Vd are both available, we apply the resistance model.

[Figure]

**Figure S16:** Sensitivity of the resistance model to uncertainties in Henry's Law coefficient (*H*) for a hypothetical compound (Y) with $H = 1 \times 10^5$ M atm$^{-1}$ (water-soluble) and $f_0 = 0$ (non-reactive). *H* was varied by several orders of magnitude for another hypothetical compound (X, $f_0$ = 0), and the percent change in $V_d$ was computed with respect to the reference $V_{d,Y}$.

The "lack of data '' in L158 refers to the lack of a reliable Henry's Law coefficient for MVK and MACR to reproduce the measurement based Vd (2.4 cm/s) by Karl et al. (2007). The resistance model estimates a Vd of roughly 0.35 cm/s, likely due to an overestimate of Rc (Fig. R2). Karl et al. (2007) suggest a much smaller (2.6 to 3.5 times) Rc for MVK and MACR. Without an accurate H, it is impossible to reproduce the Rc and thus the Vd for MVK and MACR. Therefore, we prescribe the 2.4 cm/s to MVK and MACR.

[Figure]

Fig. R2.  Left panel: the dry deposition estimated by the revised resistance model (**Base Vd**) and the prescribed Vd based on Karl et al. (**Adjusted base Vd**). Dashed lines in the left panel denote the Vd scaled by leaf area fraction and leaf angles. Right panel: leaf area distribution at the study site.

In response to this comment, we revised the text as below to clarify (lines 154-157; 161-163):

"The revised resistance model is highly sensitive to the Henry's Law coefficients (Nguyen et al., 2015) which are unknown for many species. Here, we apply the revised resistance model to species whose Henry's Law coefficients are available and measurement based Vd exist to validate the estimates, including…."

"Note that the revised dry deposition scheme yields a much smaller Vd (0.35 cm s$^{-1}$) for methyl vinyl ketone (MVK) and methacrolein (MACR) than the observation-based estimate (up to 2.4 cm s$^{-1}$, Karl et al., 2010). Therefore, we prescribe the observation-based 2.4 cm s$^{-1}$ for MVK and MACR in FORCAsT. "

P7, L159-160: "The estimates of the dry deposition velocity are then scaled by the leaf area distribution in the canopy." Does the "estimates" mean the estimate for methyl vinyl ketone and methacrolein or all the compounds in Table 2?

Dry deposition is calculated for all compounds (both those in Table 2 and other compounds) and scaled by the leaf area distribution in the canopy layers. The reason for the scaling is that the resistance model treats each canopy layer as one "big leaf" perpendicular to the sunlight. Therefore, we scale each layer by leaf area fraction and leaf angles to derive the vertical profiles of dry deposition velocity.

In response to this comment, we revised the sentence for clarification (lines 163-165):

"Because the resistance model assumes that each canopy layer is one "big leaf" perpendicular to the sunlight, the estimates of dry deposition velocities for all compounds are then scaled by the leaf area distribution in the canopy."

Section 3.3: Why the modelled O3 concentration did not show a diurnal pattern as the observation in Fig. 4? Would a spin-up run be helpful to reduce the initial peaks of ROx and NO2?

We typically do include a 12-hour spin-up on FORCaST simulations, however, they were not included in these simulations. We add 24-hour spin-up, which helps to reduce the NO2 and ROx biases (revised Fig. R3i). The discussion of RO2 is revised accordingly (Fig. R4 and Fig. R5; lines 372-381).

[Figure]

Fig. R3 (Revised Fig. 4 in the manuscript). Comparison of FORCaST 2.0 with the RCIM (**red**) and CACM1.0 (**blue**) mechanisms against observations in and above the canopy.

[Figure]

Fig. R4. The compositions of the RCIM-RO2 and CACM-RO2 during the daytime (**upper** panels) and at night (**lower** panels). Note **'Others'** in the legend refers to the RO2 from d-limonene products (such as limonaldehyde and limona ketone) and is considered as MT-RO2 in the manuscript.

[Figure]

Fig. R5. The MT-RO2 in RCIM grouped by the oxidant.

However, the spin-up does not have a large impact on the diurnal patterns of $O_3$, and the simulated diurnal cycle is still weak compared to the observed diurnal cycle of O3 (Fig. R3f) with an underestimate of O3 at night. Note that the observed $O_3$ is at the trunk space (~6 m; Table 4) and this suggests that the model may be missing some nighttime losses of O3, including mixing, dry deposition, and chemistry of NO2-NO3-N2O5 cycle and reactions with monoterpenes. In addition, horizontal advection and background O3 concentrations could also impact the O3 concentration (e.g., above and below changes in O3 are noted in Bui et al. 2021), however, we do not advect ozone in the model. As with many chemical mechanisms, it can be challenging to create large changes in O3 in remote locations While the model evaluation is not perfect, the NOx and O3 are satisfactorily reproduced by the model.

In response to this comment, we added a 24h spin-up to the model and revised Figure 4. The text is modified accordingly (line 271; 372-381):

"The model spin-up time is 24 hours."

"RCIM predicts higher ROx (HO2+ RO2) than CACM1.0 (Fig. 4i). Because both mechanisms predict similar HO2 which is discussed in the previous paragraph, here we focus on RO2 that is predominantly composed of ISOPOO and monoterpene-derived RO2(MT-RO2). Daytime (10-16 and 34-40 model hours) RCIM-RO2 is higher than CACM1.0-RO2 mainly due to differences in ISOPOO (Fig. 5a), accounting for 60% and 40% of daytime RO2 in RCIM and CACM1.0, respectively. Nighttime (0-6 and 24-30 model hours) RCIM-RO2 comprises 16% ISOPOO and 84% MT-RO2. Majority (87%) of the nighttime MT-RO2 is from NO3-initiated reactions and the remaining (13%) is O3-initiated. The observed NO3 is below the limit of detection of the instrument (1.4 pptv) on the two simulated days (not shown). While simulated nighttime NO3 is <1.4pptv (roughly 0.4 pptv) and monoterpenes agree with observations, the nighttime RO2 is still overestimated, suggesting missing sinks for nighttime RO2. "

**References**

Nguyen, T. B., Crounse, J. D., Teng, A. P., Clair, J. M. S., Paulot, F., Wolfe, G. M., and Wennberg, P. O.: Rapid deposition of oxidized biogenic compounds to a temperate forest, Proceedings of the National Academy of Sciences, 112, E392–E401. https://doi.org/10.1073/pnas.1418702112, 2015.

Karl, T., Harley, P., Emmons, L., Thornton, B., Guenther, A., Basu, C., Turnipseed, A., and Jardine, K.: Efficient atmospheric cleansing of oxidized organic trace gases by vegetation, Science, 330, 816–819, https://doi.org/10.1126/science.1192534, 2010

Bui, A. A. T., Wallace, H. W., Kavassalis, S., Alwe, H. D., Flynn, J. H., Erickson, M. H., Alvarez, S., Millet, D. B., Steiner, A. L., and Griffin, R. J.: Transport-driven aerosol differences above and below the canopy of a mixed deciduous forest, Atmos. Chem. Phys. Discuss. [preprint], https://doi.org/10.5194/acp-2021-384, in review, 2021.

**Responses to Reviewer#2**

In this work, the authors present a description of a new version of the FORCAsT canopy chemistry column model. The manuscript is well-written and is an appropriate fit for GMD. Although the chemistry included in the model is state-of-the-science, other aspects (in particular the vertical turbulent transport) are relatively standard treatments. This reviewer, having created similar models in the past, believes the time has come to move beyond these simple column models to ones that can be more useful to answering critical science questions regarding surface-atmosphere exchange of trace chemical species (gases and particles). However, documenting FORCAsT 2.0 in its current form is a valid manuscript for publication in GMD. I recommend publishing as is.

The authors thank the reviewer for the comments. We agree the turbulent transport in FORCAsT will not fully capture turbulent transport in a forest and is less sophisticated compared to models such as LES. However, there is still utility for column models to answer specific science questions regarding atmospheric chemistry in a forest canopy. Detailed gas-phase chemistry is critically important for an accurate estimation of SOA formation. While models such as LES have high vertical resolution and are useful in terms of atmosphere-biosphere exchanges, most LES use simplified chemistry that can impact the accuracy of SOA estimation and are less useful in informing regional or global models of important processes regarding SOA simulation. On the other hand, column models couple detailed chemistry with canopy processes, which enables us to investigate in-canopy chemistry especially regarding SOA formation. Perhaps the time for column models is coming to a close as the reviewer suggests, and we would welcome model developments that can accurately simulate the detailed chemistry to be able to address the science questions posed in this manuscript.